# The Study of the Sterilization of the Indoor Air in Hospital/Clinic Rooms by Using the Electron Wind Generator

**DOI:** 10.3390/ijerph16244935

**Published:** 2019-12-05

**Authors:** Józef S. Pastuszka, Walter Mucha, Agnieszka Wlazło, Danuta Lis, Ewa Marchwińska-Wyrwał, Anna Mainka

**Affiliations:** 1Faculty of Energy and Environmental Protection, Department of Air Protection, Silesian University of Technology, 44-100 Gliwice, Poland; Jozef.Pastuszka@polsl.pl (J.S.P.); Walter.Mucha@polsl.pl (W.M.); 2Institute of Occupational Medicine & Environmental Health, 41-200 Sosnowiec, Poland; a.wlazlo@imp.sosnowiec.pl (A.W.); d.lis@imp.sosnowiec.pl (D.L.); 3Faculty of Public Health, Silesian University of Medicine, 41-902 Bytom, Poland; e.marchwinska@wp.pl

**Keywords:** electron wind generator (EWG), bacteria, fungi, hospitals, patient room, waiting room, ozone, bacteria size-distribution, fungi size-distribution

## Abstract

(1) Background: Since exposure to airborne bacteria and fungi may be especially hazardous in hospitals and outpatient clinics, it is essential to sterilize the air in such rooms. The purpose of this study was to estimate the decrease in the concentration of airborne bacteria and fungi in the selected hospital and clinic rooms due to the work of the electron wind generator (EWG). (2) Methods: EWG is an air movement and air purification device using a sophisticated combination of electrode topology and specially designed high-voltage power supply. (3) Results: The concentration of both bacteria and fungi in the small patient’s room dropped to approximately 25% of the initial (background) concentration. In the larger patient’s room, the concentration dropped to 50% and 80% of the background concentration for bacteria and fungi, respectively. (4) Conclusions: The obtained data show that the studied sterilization process can be described by the exponential function of time. Moreover, the application of an activated carbon filter into EWG significantly decreases the concentration of ozone in the sterilized room. Sterilization by EWG significantly changes the characteristic of species and genera of airborne bacteria and shifts the main peak of the size distribution of airborne bacteria into the coarser bio-particles.

## 1. Introduction

Bioaerosols are a loosely defined group of airborne particles of biological origin, generally including bacteria, fungi, and viruses, as well as pollen, their fragments, and various antigens. They can cause many adverse health effects, including allergic, toxic, and infection responses [1,2,3,4]. Since people spend the most time in the indoor environment, concentrations of airborne bacteria and fungi indoors have been intensively studied during the last two decades [5,6,7,8,9,10,11]. Exposure to bioaerosols may be especially hazardous in clinics and hospitals. Some bacteria such as *Streptococcus pyogenes, Neisseria meningitides, Corynebacterium diphtheria,* and *Mycobacterium tuberculosis* are known to be transmitted predominantly by airborne droplets from infected people, and they may cause nosocomial infection [12,13]; however, the room’s airflow patterns also play a significant role in bioaerosol transport [14].

Therefore, it is very important to clean the air in such buildings, especially in the surgery rooms. Over the years, a number of studies have been undertaken to develop air cleaning technologies. Among these technologies, filtration methods [15] have become very popular. Especially interesting seems to be the carbon nanotube filters to collect aerosolized biological and non-biological particles [16]. However, appropriate techniques should not only collect the biological particles from the air, but also inactivate them. To achieve this purpose various technologies have been attempted including the use of electrostatic field [17], as well as unipolar ion emitters and UV-enhanced TiO_2_ cells [18]. Among a number of different portable cleaners, electrostatic devices seem to be especially promising [19].

The purpose of the reported study was to estimate the decrease in the concentration of biological aerosols in the selected hospital and clinic rooms due to the work of the electron wind generator (EWG). It is an air movement and air purification device using a sophisticated combination of electrode topology and specially designed high-voltage power supply. Alternatively, this represents a simple design to allow for assembly without advanced equipment or technically skilled labor. 

Our work was also aimed to reduce ozone emission caused by the strong sparking occurring after a few hours of work of this device. Therefore, in the next stage of our study, the EWG sterilizer was equipped with a carbon filter. The first stage of our studies on the subject started in 2001–2009 and have been continued in the last ten years. The preliminary results of the first stage of our research were presented [20].

## 2. Materials and Methods

During this study, the measurements of the concentration of the bacterial and fungal aerosols were carried out in two hospitals and clinics as a function of working time of the EWG device being the portable air sterilizer. We used two EWG devices (Figure 1) from POLUS Ltd. (Bytom Polus Enterprises Ltd. was organized in the Polish–American cooperation running in 2001–2018).

Measurements were carried out in two hospitals and one medical clinic located in Sosnowiec and Zabrze, Upper Silesia, Poland. Detail information on sampling sites has been presented in Table 1.

Microorganisms were collected by using the six-stages Andersen impactor on nutrient media, specific to either fungi or bacteria, in Petri dishes located on all impactor stages. The pump provided a constant flow rate of 28.3 dm^3^ min^−1^ during measurements. Aerodynamic cut-size diameter for these stages were following: 7.0, 4.7, 3.3, 2.1, 1.1, and 0.65 µm. Sampling time following Nevalainen et al. [21] was 5–10 minutes. As a minimum, sequential duplicate samples were obtained for every measurement, at least three samples were taken. During the measurement, the Andersen impactor was placed in the center of the rooms at the height of 1.5 m while the EWG device was placed at the same height but near the closed door (at a distance of about 2 m).

Malt extract agar (MEA 2%) and trypcase soy agar (TSA) were applied for fungi and bacteria, respectively. All samples were incubated for seven days at temperature (22 °C). Concentrations of viable bacteria and fungi were calculated as colony-forming units per cubic meter of air (CFU m^−3^) using positive hole correction. In addition, for all selected rooms, two bacterial samples (background and when the decreasing concentration was beginning to stabilize) were identified. The bacterial aerosol samples were identified according to Gram staining and morphology. Next, visible colonies were sub-cultured onto either Chapman agar or onto MacConkey agar. Gram-positive and gram-negative bacteria were finally identified by the active pharmaceutical ingredients API test.

It should be noted that the culture-based method employed in this study includes probably only about 10% of the total microorganisms present in the air [22]; however it still provides valuable information for the assessment of patients’ exposure and the efficiency of the sterilization using the EWG device. Furthermore, air sampling for viable bacteria, forming colonies on the agar, can be valuable in the identification of the environmental bacterial species that have been linked specially with diseases [23].

To reduce the emission of ozone from the EWG device, this sterilizer was equipped, in the next stage of our study, with a carbon filter and also a fabric filter that would capture coarse particles. However, then it would be essential to increase the air stream that is taken in, which can be achieved by means of a small fan. 

Quantitative determination of the ozone level has been performed using the colorimetric method with dimethyl-p-phenylenediamine. The sample collection included passing 10 dm^3^ of the air at a rate of 1 dm^3^/min through a two-scrubber containing 10 cm^3^ of a 1% KI solution. The determination of ozone concentration in the sample included 5 cm^3^ of the absorbing solution transferred to a colorimetric tube, next 0.5 cm^3^ of 0.02% dimethyl-p-phenylenediamine hydrochloride was added and mixed. After 15 minutes, the color of the tested solution was compared in visible light (λ = 550 nm) to the reference prepared in accordance with the scale of standards. Ozone concentrations in the absorption solution have been measured by means of a scanning spectrophotometer UV−VIS by Shimadzu (Shimadzu Corporation, Kyoto, Japan). 

## 3. Results and Discussion

The changes in the concentration levels of viable bacteria and fungi due to the sterilization using the EWG devices in the selected hospital rooms are presented in Table 2, Table 3 and Table 4. In the studied rooms, the background concentration of bacterial aerosol ranged from 10^2^ to 10^3^ CFU m^−3^, and the concentration level of fungal aerosol was about 10^2^ CFU m^−3^. As it could be expected, the obtained results show that the concentration of viable bioaerosols depends strongly on the number of patients present in the room, capacity/volume of the room, and ventilation method.

Analysis of Table 2 indicates that after four hours of sterilization using two EWG devices, the concentration of both bacteria and fungi in the small patient’s room (PR_S_) dropped to approximately 25% of the initial (background) concentration. It is also interesting that during the sterilization, the relative humidity decreased from 65% to 54% after 4 hours. 

Table 3 shows that in the larger patient’s room (PRL) and during the weaker sterilization process (only one EWG device was used while the space of this room was larger compared to other rooms) the decrease of the concentration of bioaerosol in the hospital in Zabrze was not as concentrated as that in the hospital in Sosnowiec. However, even in this room, the concentration dropped to 50% and 80% of the background concentration for bacteria and fungi, respectively.

A significant reduction in bioaerosol levels due to the work of the EWG device was also received in the clinical waiting room (Table 4). It should be noted that after four hours of sterilization, the concentration of bacteria and fungi decreased, compared to the background level, up to about 40% and 50%, respectively. Continuation of the sterilization process in this waiting room up to 6 hours showed (Figure 2) that this process can be described by an exponential function of time: (1)$$C\left(t\right)=C\left(0\right)e^{-\mathrm{kt}},$$
where *k* defines sterilization efficiency.

Thus, for the WR room, with a specific number of people, the sterilization coefficient, *k*, can be determined as 0.23 and 0.11 h^−1^ for the airborne bacteria and fungi, respectively. Therefore, for a current or assumed concentration levels *C(0)* the sterilization time, *t*, necessary to reduce the bioaerosol concentration to an acceptable level, *C_accepted_*, can be calculated as follows:(2)$$t=\;\frac{1}{k}\ln\frac{C\left(0\right)}{C_{\mathrm{accepted}}}$$

For example, assuming that the measured concentration of bacteria in this clinical WR is 1000 CFU m^−3^ and the accepted level is 100 CFU m^−3^ the needed sterilization time is about 10 hours. Since the aerodynamic diameter of airborne particles, including bioaerosol particles, determines their behavior not only in the air but also in the human respiratory tract after inhalation, it is important to know what the size distribution of airborne bacteria and fungi in the studied indoor environment is. Such distributions are presented in Figure 3 and Figure 4 and Appendix A. As can be seen, the patterns of size distribution without any sterilization (background data) vary, depending mainly on the type of ventilation and number of persons in the room. It should be noted that the main peak in the size distribution of airborne bacteria in the mechanically ventilated hospital rooms (PR_S_) in Sosnowiec appears in the size range 3.3–4.7 µm, i.e., it was shifted into larger particles compared to the size distribution in the hospital in Zabrze (PR_L_) and the outpatient clinic in Sosnowiec (WR) where there was only natural ventilation. A similar, but not so clear relationship can be seen for the airborne fungi. This result indicates that biological particles attached to relatively coarse dust particles can be emitted from the ventilation ducts. Resuspension/reemission process, of coarser particles of bacteria present in the settled dust, generated by air blowing from heating, ventilation, and air conditioning system HVAC, is also possible.

The similar shift of the main peak into the coarser bio-particles can be seen in the shape of the curves of the size distribution of airborne bacteria after four hours of sterilization compared to the size distribution before the sterilization. Certainly, it is the consequence of the fact that the finest airborne bacteria can be attracted to the EWG sterilizer easier than coarse, heavy particles. As a result, the contribution of fine bioaerosol particles to the total amount of airborne bacteria and fungi decreases after the sterilization. However, in the mechanically ventilated room, the concentration of bioaerosol is rather low, especially the concentration of fine particles (Table 2). As a result, the ratio of the concentration of respirable bacterial particles, having aerodynamic diameter up to 4.7 µm, to the total airborne bacteria decreased only little; from 0.45 to 0.43. On the other hand, in the hospital in Zabrze and the outpatient clinic (WR) without mechanical ventilation, this decrease was from 0.85 to 0.78 and from 0.72 to 0.53, respectively (Table 3 and Table 4).

The efficiency of the sterilization of fungi was much less than of bacteria (see Table 2, Table 3 and Table 4). It confirms the general statement that airborne bacteria are much more sensitive to the influence of different physical and chemical stress than airborne fungi are. Important information about the changing of the exposure to bioaerosol due to the work of the EWG device can be obtained from the detailed analysis of the bacterial genera and species. The general pattern of these changes in the viable bacteria genera and species due to the sterilization was the same for all studied indoor environments and can be illustrated, as an example, for the patients’ room (PR_L_) at the University Hospital in Zabrze (Table 5). As Table 5 shows, the dominant group of bacteria isolated from the studied hospital air was *Staphylococcus/Micrococcus*. The main source of these bacteria indoors is the human organism, especially human skin [24]. It can be seen that gram-positive cocci constituted 77% of all detected airborne bacteria. Non-sporing gram-positive rods, including mainly *Corynebacterium*, contributed 19% of the total bacterial flora. It should be noted that in the hospital in Zabrze some actinomycetes were found. 

After 4 hours of sterilization, the gram-positive cocci had become even more dominant constituting 90% of the total bacteria. Probably there were two reasons for this result. First, only these bacteria genera/species are currently and significantly emitted from patient’s organisms during the sterilization. Second, these microorganisms due to their spherical shape can be destroyed more effectively than rods. 

During the measurements, all patients were in their rooms lying or sitting on the beds. Some went out to the toilet for about 15 minutes. On average, two to three patients went to the toilet during 4–5 hour measurement. During the research, no one entered the patients’ rooms.

During our measurements in the hospital in Sosnowiec, all windows were closed. Unfortunately, in the old hospital in Zabrze windows were frequently open although we asked patients to keep the windows closed. It can be estimated that during the 4 hours one window was open three times for about 10–15 minutes (in every case different window).

Although the efficiency of the sterilization of the indoor air in the studied hospitals and clinic by the EWG was acceptable for their managers, unfortunately, the high voltage used in this device generated the significant emission of ozone. Therefore, in the next step of our study, we applied the carbon filter in the EWG to reduce the emission of ozone. Table 6 shows the concentration of ozone in the naturally ventilated office room of 85 m^−3^ (OR_L_) as a function of sterilization time for the EWG with and without the carbon filter. As can be seen, the concentration of ozone emitted from the modernized hybrid EWG device was about three times lower compared to the ozone released from this sterilization without a carbon filter. 

The measurements of the concentration of airborne bacteria carried out in the small office room of 40 m^−3^ (OR_S_) as a function of time counted from the beginning of sterilization using the new hybrid cleaner showed (Table 7) that the efficiency of the sterilization for the modernized EWG device is still adequate.

## 4. Conclusions

These studies were oriented toward the estimation of the decrease in the concentration of bacterial and fungal aerosol in the selected hospital and outpatient clinic rooms due to the work of the portable electron wind generator (EWG). It is an air movement and air purification device using a sophisticated combination of electrode topology and specially designed high-voltage power supply.

The best results were obtained in small rooms, e.g., in a three-to-four patient room in the hospital (PR_S_). In such rooms, with only sporadically opened windows, the concentration of bacterial and fungal aerosol after four hours of EWG operation dropped to 24% and 25% of the previous, background level, respectively. In the larger patient’s room (PR_L_) or/and during the weaker sterilization process the concentration dropped to 50% and 80% of the background concentration for bacteria and fungi, respectively. 

It has been showed that the sterilization process can be described by the exponential function of time where sterilization efficiency, *k*, was 0.23 and 0.11 h^−1^, for bacteria and fungi, respectively. The studied sterilization process significantly changed the character of species and genera of airborne bacteria. After four hours of sterilization, the gram-positive cocci constituted 90% of the total bacteria. After four hours of sterilization by the EWG device, the main peak in the size distribution of airborne bacteria was shifted into the coarser bio-particles, which implies that fine particles are more easily separated from the air. 

The efficiency of sterilization of fungi was much less than that of bacteria confirming the known statement that airborne bacteria are much more sensitive to the influence of different physical and chemical stress than airborne fungi are. Anyway, similar, but not as clear relationships as for bacteria, were also obtained for the fungal aerosol.

The application of an activated carbon filter into a portable, high-voltage air sterilizer (EWG) cuts the emission of ozone from this appliance down by 60%–80%. Such a significant fall of emission leads to ozone concentration in air to being reduced about three times, after several hours of operation of the sterilizer equipped with a carbon filter in a naturally ventilated office room (OR_L_) of 85 m^3^, compared to the concentration of ozone which is produced when the sterilizer is working without the carbon filter. The tested carbon filter guarantees the reduction of ozone emission during the continuous working of the studied device for up to six hours. During the long term of work, a filter with a higher mass density of activated carbon needs to be used.

The possible application of the modernized hybrid EWG devices in the hospitals and medical clinics seems to be very promising, as the proposed method of reducing bioaerosols in the healthcare environment does not require occupants to leave the room. 

## Figures and Tables

**Figure 1 ijerph-16-04935-f001:**
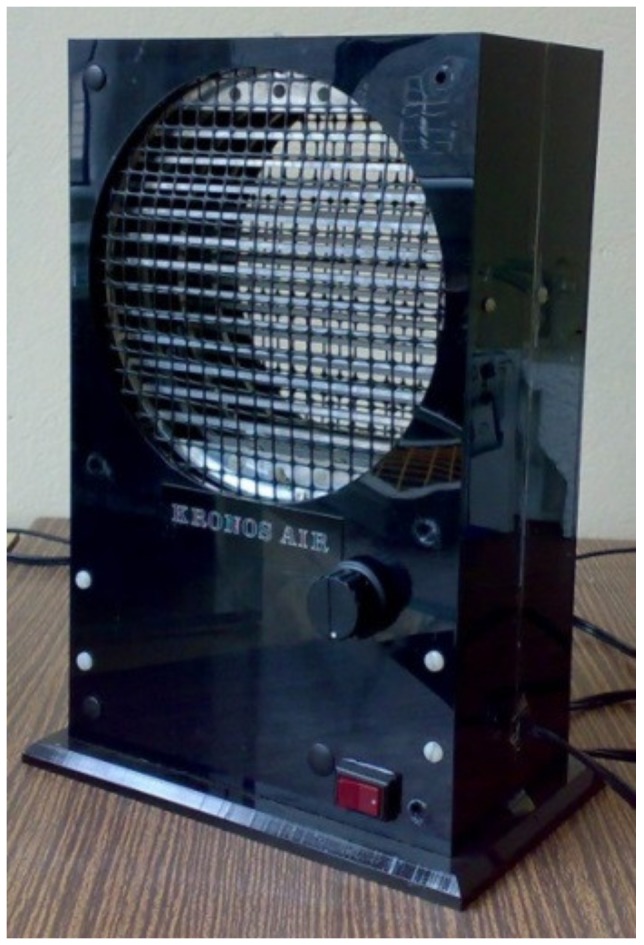
Photograph of the electron wind generator (EWG) device.

**Figure 2 ijerph-16-04935-f002:**
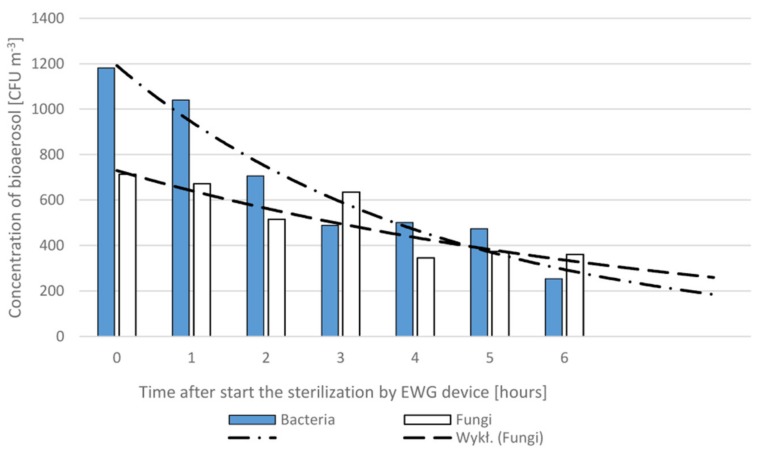
Long-time decrease of the concentration level of airborne bacteria and fungi in the waiting room at the clinic in Sosnowiec due to the air sterilization using the EWG device.

**Figure 3 ijerph-16-04935-f003:**
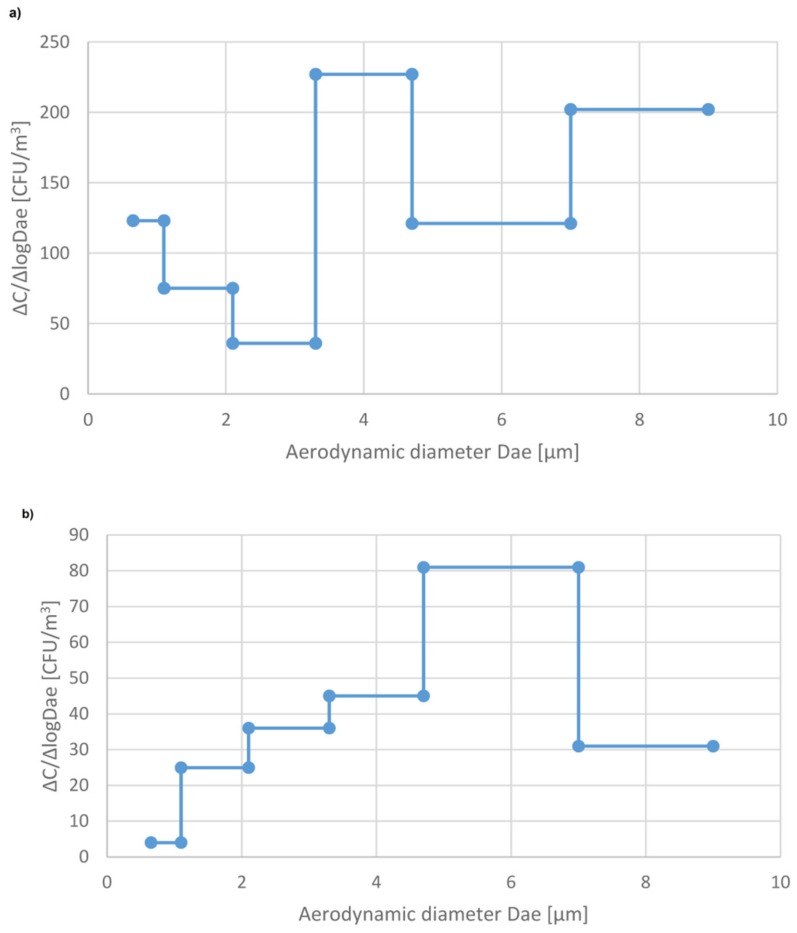
Size distribution of airborne bacteria in the patient’s room (PR_S_) at the hospital in Sosnowiec: (**a**) background, (**b**) 4 hours after the start of sterilization using the EWG device.

**Figure 4 ijerph-16-04935-f004:**
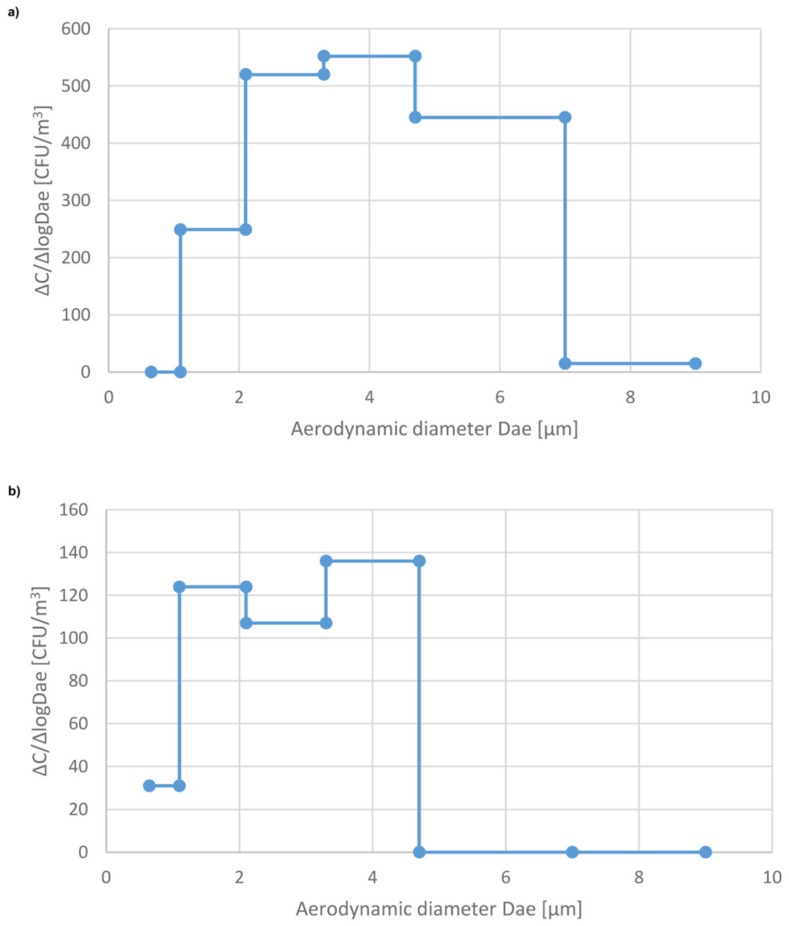
Size distribution of airborne fungi in the patient’s room (PRS) at the Hospital in Sosnowiec: (**a**) Background, (**b**) 4 hours after the start of sterilization using the EWG device.

**Table 1 ijerph-16-04935-t001:** Characteristics of sampling sites in hospitals.

Sampling Site	City	Volume, m^3^	No. of Patients	Temperature Indoors, °C	Relative Humidity, %	Ventilation	Windows
Patient room small (PR_S_)	Sosnowiec	30	4–5	24	65–54	mechanical	closed
Patient room large (PR_L_)	Zabrze	110	8	23	79–80	natural	opened
Waiting room (WR)	Sosnowiec	77	4–13 *	22	62–66	natural	closed

* Plus 2–3 persons crossing this room.

**Table 2 ijerph-16-04935-t002:** Changes in the concentration level of bacterial and fungal aerosol in the small patient’s room (PR_S_) at the hospital in Sosnowiec due to the sterilization using one EWG device.

Concentration of Viable Microorganisms in the Room [CFU m^−3^]						
**Parameter**	**Bacteria**			**Fungi**		
	Background	2 h	4 h	Background	2 h	4 h
C_Total_	204	63	49	341	261	84
C_Resp_	91	42	21	257	261	84
C_Resp_/C_Total_	0.45	0.67	0.43	0.75	1.00	1.00
RH, %	65	58	54	65	58	54
C^4 hours^ _Total_/C^Background^ _Total_	0.24			0.25		

Resp. particles with d < 4.7 μm. CFU, colony-forming units.

**Table 3 ijerph-16-04935-t003:** Changes in the concentration level of bacterial and fungal aerosol in the large patient’s room (PR_L_) at the University Hospital in Zabrze due to the sterilization using one EWG device.

Concentration of Viable Microorganisms in the Room [CFU m^−3^]						
**Parameter**	**Bacteria**			**Fungi**		
	Background	2 h	4 h	Background	2 h	4 h
C_Total_	848	452	445	872	854	698
C_Resp_	721	339	346	784	763	635
C_Resp_/C_Total_	0.85	0.75	0.78	0.90	0.89	0.91
RH, %	80	79	80	80	79	80
C^4 hours^ _Total_/C^Background^ _Total_	0.52			0.80		

Resp. particles with d < 4.7 μm.

**Table 4 ijerph-16-04935-t004:** Changes in the concentration level of bacterial and fungal aerosol in the waiting room (outpatient room), WR, at the clinic of occupational diseases in Sosnowiec due to the sterilization using one EWG device.

	Concentration of Viable Microorganisms in the Room [CFU m^−3^]							
**Parameter**	**Bacteria**				**Fungi**			
	Background	1 h	2 h	4 h	Background	1 h	2 h	4 h
C_Total_	1181	1040	706	501	713	671	515	345
C_Resp_	849	927	451	268	650	622	459	268
C_Resp_/C_Total_	0.72	0.89	0.64	0.53	0.91	0.93	0.89	0.78
RH, %	75	66	64	62	75	66	64	62
C^4 hours^ _Total_/C^Background^ _Total_				0.42				0.48

Resp. particles with d < 4.7 μm.

**Table 5 ijerph-16-04935-t005:** Changes in the viable bacteria genera in the patients’ room (PR_L_) at the University Hospital in Zabrze, after 4 hours of sterilization by using the EWG cleaner (two cleaners worked, started at 10:00 after measuring the “background” concentration).

Bacteria Species and Genera	Background		4 Hours of Sterilization *	
	[CFU m^−3^]	[%]	[CFU m^−3^]	[%]
Gram-positive cocci, including:	573	77	381	90
*Staph. epidermidis*	198		0	
*Staph. saprophyticus*	74		148	
*Staph. hominis*	11		14	
*Staph. xylosus*	85		0	
*Staph. cohnii cohnii*	11		7	
*Micrococcus* spp.	166		191	
*Kocuria*	28		21	
Non-sporing gram-positive rods, including:	140	19	28	7
*Corynebacterium striatum/amycolatum*	75		0	
*Corynebacterium aquaticum*	25		7	
*Corynebacterium propinquum*	25		7	
*Corynebacterium pseudodiphtheriticum*	4		0	
*Microbacterium* spp.	11		0	
*Brevibacterium* spp.	0		14	
Actinomycetes, including:	33		14	3
*Rhodococcus* spp.	21		14	
Others	12	4	0	
Total	746	100	423	100

* During 4 hour measurements, two patients went to the toilet (not simultaneously). One patient returned after 15 minutes and the other after about 20 minutes.

**Table 6 ijerph-16-04935-t006:** Time dependence of the concentration of ozone in the office room (OR_L_) sterilized by using the EWG device.

Concentration of Ozone [µg m^−3^]	Sterilization Time [h]			
	0	2	4	6
	(Background)			
without carbon filter	0.50	2.65	3.15	3.25
with carbon filter	0.20	0.60	1.20	1.25

**Table 7 ijerph-16-04935-t007:** Changes in the concentration level of bacterial aerosol in the office room (OR_S_) due to the sterilization using one EWG device having a carbon filter.

Parameter	Concentration of Viable Bacteria in the Room [CFU m^−3^]			
	0	1	6	9
	(Background)	(Time, in Hours, Counted from the Beginning of Sterilization)		
C_Total_	1293	1166	385	293
C^n hours^/C^Background^	1.00	0.90	0.30	0.23

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
