# Peer review of "The Study of the Sterilization of the Indoor Air in Hospital/Clinic Rooms by Using the Electron Wind Generator"

_ijerph, 2019, doi:10.3390/ijerph16244935_

Round 1
Reviewer 1 Report
L64 – no need to say it has been presented at a conference
L69 – no need to say rented
L82 – specify the room temperature
Table 2 – need to specify how many samples were taken and replicates (possibly earlier in method)
Fig 1 – spelling of bioaerosol on y axis
Fig 2 – no comment to make on the concentrations of particles as well as size?
L153-163 – Settling rates of the various particle sizes needs consideration, would larger coarser particles be likely to settle out more rapidly? This is why a discussion on what disturbance was occurring in the room is needed as it will impact on particles aerosolised.
L177-178 – Gram-negative bacteria don’t aerosolise that well in any event, and additionally because they could not be cultured does not mean they were not there. This is a limitation that is not discussed.
L179-180 – not necessarily as Actinomycetes are fairly ubiquitous.
Table 5 – the differences could as easily be accounted for by general variation, which is why we need more information on replicates and room activity.
L190 – doesn’t make sense English is poor
L202 – English is colloquial and needs to be more academic.
Comment needs to be made about activity in and around the room, did people come and go? Were patients there the whole time? Did doors open and close?
Because we lack information on the methodology it is not clear the conclusions can be supported. The queries raised need to be clarified prior to publication. More discussion is also needed on the limitations of sampling and enumerating culturable bioaerosols.
Author Response
November 25th, 2019
Title: "The study of the sterilization of the indoor air in hospital / clinic rooms by using the electron wind generator"
Manuscript ID: ijerph-637282
Dear Reviewer 1,
Thank you for your message with comments on our manuscript. We appreciate the responses and helpful suggestions for improving the presentation of our study. We have followed the comments and marked them in the text (red colour).
L64 – no need to say it has been presented at a conference
Corrected into: The preliminary results of the first stage of our research were presented in Karlsruhe [20].
L69 – no need to say rented
Corrected into:
We used two EWG devices (Figure 1) from the POLUS Ltd. (Bytom Polus Enterprises Ltd. was organized in the Polish–American cooperation running in 2001-2018).
L82 – specify the room temperature
This sentence has been changed as follows:
All samples were incubated for 7 days at the temperature 22oC.
Table 2 – need to specify how many samples were taken and replicates (possibly earlier in method)
As a minimum, sequential duplicate samples were obtained for every measurement. In most cases 3 samples were taken.
Fig 1 – spelling of bioaerosol on y axis
Corrected
Fig 2 – no comment to make on the concentrations of particles as well as size?
Brief discussion on the data presented in Fig. 2 is on page 5 (line 150-158). Information about the concentration levels can be found in Table 2.
L153-163 – Settling rates of the various particle sizes needs consideration, would larger coarser particles be likely to settle out more rapidly? This is why a discussion on what disturbance was occurring in the room is needed as it will impact on particles aerosolised.
When bioaerosol particles fall or are drawn (or propelled) through air by gravitational force or an electric field, their velocity increases or decreases from some initial value until the atmospheric drag force is equal to the force field, causing it to move. In the case of gravitational force, this is the terminal settling velocity of the particle. For example the terminal settling velocity for virus size particles may be approx. nil (<10-9 cm s-1), whereas 100 mm agglomerations or rafts may settle at 25 cm s-1 [1]. Dispersal patterns of spray nozzle fractions containing large, presumably nonevaporated droplets of ³ 7 mm in aerodynamic diameter, and two smaller fractions, one with diameters of 2-3 mm (probably the residue of droplets containing more than one spore) and 1-2 mm in diameter (probably the residue from single-spore droplets) showed that progressively larger droplet residues settle to progressively lower heights [1].
It is reasoned that the large particles are of local origin because an 8 mm diameter particle has a sedimentation velocity in still air of about 0.20 cm s-1, and a 3 mm particle has a sedimentation velocity in still air of 0.002 cm s-1. Thus, an 8 mm particle would take about 15 min to settle 2 m, whereas a 3.3 mm particle would take over a day [1].
L177-178 – Gram-negative bacteria don’t aerosolise that well in any event, and additionally because they could not be cultured does not mean they were not there. This is a limitation that is not discussed.
We can only partially agree with this opinion. Literature data showing migration of Gram-negative bacteria from soil to the indoor environment can be found. There are documented data indicating that samples of Gram-negative bacteria developed on an agar and could be precisely classified.
L179-180 – not necessarily as Actinomycetes are fairly ubiquitous.
We agree. We deleted the second part of this sentence: “It should be noted that in the hospital in Zabrze some actinomycetes were found. what indicates moisture problems in the building.”
Table 5 – the differences could as easily be accounted for by general variation, which is why we need more information on replicates and room activity.
Although we took two air samples before sterilization and two samples after four hours, we performed species composition analysis for only one background sample and one sample after 4 hours of sterilization. However, species analysis of other samples carried out in Sosnowiec also showed an increase in Gram-positive cocci content in the air due to sterilization.
We also added the information below Table 5:
During 4-hour measurements, 2 patients went to the toilet (not simultaneously). One patient returned after 15 minutes and the other after about 20 minutes.
L190 – doesn’t make sense English is poor
Corrected into: …by the EWG was acceptable for their managers, unfortunately, the high voltage used…
L202 – English is colloquial and needs to be more academic.
“satisfying” was corrected into “adequate”
Comment needs to be made about activity in and around the room, did people come and go? Were patients there the whole time? Did doors open and close?
During the measurements, all patients were in their rooms lying or sitting on the beds. Some went out to the toilet for about 15 minutes. On average, 2-3 patients went to the toilet during 4-5 hour measurements. During the research, no one entered the patients' rooms.
During our measurements in the hospital in Sosnowiec all windows were closed. Unfortunately, in the old hospital in Zabrze windows were frequently open although we asked patients to keep the windows closed. It can be estimated that during 4 hours one window was open 3 times for about 10-15 minutes (in every case different window).
Because we lack information on the methodology it is not clear the conclusions can be supported. The queries raised need to be clarified prior to publication. More discussion is also needed on the limitations of sampling and enumerating culturable bioaerosols.
The bacterial aerosol samples were identified according to Gram staining and morphology. Next, visible colonies were subcultured onto either Chapman agar or onto MacConkey agar. Gram-positive and Gram-negative bacteria were finally identified by the biochemical API test.
It should be noted that the culture-based method employed in this study include probably only about 10% of the total microorganisms present in air (Li et al., 2017); however it still provides valuable information for the assessment of patients` exposure and the efficiency of the sterilization using EWG device. Besides, air sampling for viable bacteria, forming colonies on the agar, can be valuable in the identification of the environmental bacterial species that have been linked specially with diseases (An AIHA Biosafety Guide, 1996).
References
Li Y., Lu R., Li W.,Xie Z., Song Y. (2017) Concentrations and size distributions of viable bioaerosols under various weather conditions in a typical semi-arid city of Northwest China. J. Aerosol Sci. 106, 83-92.
A Publication of the American Industrial Hygiene Association (1996) Field Guide for the Determination of Biological Contaminants in Environmental Samples, AIHA, Fairfax, VA.
Again, we would like to express our appreciation to you and the reviewers for your efforts and helpful comments. Please find the revised version of our paper enclosed.
Yours sincerely,
Anna Mainka
Reviewer 2 Report
I have read carefully the manuscript “The study of the sterilization of the indoor air in hospital/clinic rooms by using the electron wind generator” by Józef S. Pastuszka et al. Overall the manuscript is written clearly and experiments has been well planned. The authors estimated the efficiency of Electron Wind Generator (EWG) in reducing the number of airborne microorganisms in various hospital rooms. The problem of bacterial/fungal contamination in hospitals is very important and any methods leading to reduce the risk are highly desirable. Therefore, I think the presented manuscript can be considered for publication. However, I have found some points that need to be explained and improved.
In my opinion, the information concerning the EWG device is too superficial. There is only one sentence describing it: “It is an air movement and air purification device using a sophisticated combination of electrode topology and specially designed high voltage power supply. On the other hand, this represents a simple design to allow for assembly without advanced equipment or technically skilled labor”. I could not find the POLUS company. What is EWG purifier, and how does it work? There are a lot of commercial air purifiers on the market that use sophisticated multi-stage filtration including electrostatic one. The authors did not present any photo of the device and even reference to the source where the reader could find more information. It would be interesting to know the device‘s working principle. Does it use high voltage only to destroy microorganisms or has an additional HEPA filter? What is the airflow rate? I am not convinced that “Electron Wind Generator” properly describes the working principle of the device. It is confusing with wind turbines. At least this keyword refers to electric power production. Please, precise the “mechanical ventilation” in Patient room in Sosnowiec Hospital. Does it use any filtration? Is the air vacuumed or pressed to the room? Why the particle distribution pattern in fig 2a consist so many large particles? The explanation in L148 that “mechanical ventilation is more effective in removing fine particles than coarse particles” can be also explained by air movement that makes them flying. Please explain where the POLUS was placed in the room during experiments?. In the middle? Where was the Andersen sampler placed? Please give any impression on the room volume or the ratio between rooms. Please use the same tics in Fig 2 b as in rest of figures. L79-80 what was the airflow rate used for Andersen impactor. L82 – Please write whole procedure of bacteria culturing or give a reference to the procedure. Why the bacteria were grown at room temperature? The human pathogens has an optimum at 37C. L91-94 – please improve the description of ozone capturing for measurement. What was the absorption wavelength used. How was the calibration curve done? L112 – “…and during the weaker sterilization..” – What did you mean? Did you use different voltage setup that influenced the sterilization intensity? – please improve L120 – “Significantly lower results…” a) if lower bacteria/fungi concentration than I do not agree. B) may be lower sterilization efficiency.. – please improve. Moreover, significantly lower are results that are presented in Tab 3 than in Tab 4, comparing to those in Tab 2. In figure 1 authors proved exp. decay in WR in Sosnowiec. What about two remaining rooms? Did they confirm the theory? Please refer to other cases.Author Response
November 25th, 2019
Title: "The study of the sterilization of the indoor air in hospital / clinic rooms by using the electron wind generator"
Manuscript ID: ijerph-637282
Dear Reviewer 2,
Thank you for your message with comments on our manuscript. We appreciate the responses and helpful suggestions for improving the presentation of our study. We have followed the comments and marked them in the text (blue colour).
In my opinion, the information concerning the EWG device is too superficial. There is only one sentence describing it: “It is an air movement and air purification device using a sophisticated combination of electrode topology and specially designed high voltage power supply. On the other hand, this represents a simple design to allow for assembly without advanced equipment or technically skilled labour”.
We have completed the information on the EWG device.
I could not find the POLUS company.
The POLUS Ltd. (Bytom Polus Enterprises Ltd. was organized in the Polish–American cooperation running in 2001-2018).
What is EWG purifier, and how does it work?
Under an electrostatic field surrounding the electrode wire at a high voltage relative to ground potential gas molecules, particles and bioaerosols are charged and attracted to an oppositely charged electrode.
There are a lot of commercial air purifiers on the market that use sophisticated multi-stage filtration including electrostatic one. The authors did not present any photo of the device and even reference to the source where the reader could find more information. It would be interesting to know the device‘s working principle.
The scheme of electrodes topology had been restricted by the company, at the beginning of our research. However, we can add a photograph of the device (Figure 1).
Does it use high voltage only to destroy microorganisms or has an additional HEPA filter?
At the first stage of study, only EWG powered by high voltage was used. No filters were used during the measurements in the hospitals and clinic.
At the second stage a fabric/car air filter was used before EWG and carbon car air filter was used after EWG, but HEPA filter was not used in the construction. The results of this study are presented in Tables 6 and 7 only.
What is the airflow rate?
The EWG airflow forced by air ionization was 41 m3/h, the use of air filters reduced the airflow almost completely, so there was a need to use the ventilator with the maximum flow of 148 m3/h.
I am not convinced that “Electron Wind Generator” properly describes the working principle of the device. It is confusing with wind turbines. At least this keyword refers to electric power production.
We agree that “Electron Wind Generator” is not an ideal name however, this name has been proposed by the company.
Please, precise the “mechanical ventilation” in the Patient room in Sosnowiec Hospital. Does it use any filtration? Is the air vacuumed or pressed to the room?
A central mechanical ventilation system (HVAC) was installed in the hospital building in Sosnowiec, but we were not able to assess its effectiveness. We did not have access to the HVAC documentation, so it is not possible to provide any detailed information on this subject.
Why the particle distribution pattern in fig 2a consist so many large particles?
This is a very valuable question. We observed similar size distributions in some buildings with old or long-time uncleaned ventilation systems (HVAC), which are switched on only periodically (they do not work continuously). It seems that biological particles attached to relatively coarse dust particles are emitted from the ventilation ducts. Resuspension/reemission process, of coarser particles of bacteria present in the settled dust, generated by air blowing from HVAC, is also possible.
The explanation in L148 that “mechanical ventilation is more effective in removing fine particles than coarse particles” can be also explained by air movement that makes them flying.
We agree. The sentence “Such a result indicates that mechanical ventilation is much more effective in removing fine particles than coarse particles” will be replaced by the following explanation:
“This result indicates that biological particles attached to relatively coarse dust particles can be emitted from the ventilation ducts. Resuspension/reemission process, of coarser particles of bacteria present in the settled dust, generated by air blowing from HVAC, is also possible.”
Please explain where the POLUS was placed in the room during experiments?. In the middle?
Where was the Andersen sampler placed?
During the measurement the Andersen impactor was places in the center of the rooms at a height of 1.5 m . The EWG device was placed at the same height but near the closed door (at a distance of about 2 m).
Please give any impression on the room volume or the ratio between rooms.
The volume of the rooms is in Table 2.
Please use the same tics in Fig 2 b as in rest of figures.
The figures have been corrected.
L79-80 what was the airflow rate used for Andersen impactor.
Pump provided a constant flow rate of 28.3 dm3 min-1 during measurements.
L82 – Please write whole procedure of bacteria culturing or give a reference to the procedure.
The bacterial aerosol samples were identified according to Gram staining and morphology. Next, visible colonies were subcultured onto either Chapman agar or onto MacConkey agar. Gram-positive and Gram-negative bacteria were finally identified by the biochemical API test.
Why the bacteria were grown at room temperature? The human pathogens has an optimum at 37C.
It is true: for the isolation of human pathogenic organisms, plates should be incubated at 37oC. However, incubation at lower temperatures may recover a greater number of species and give improved resuscitations of stresses bacteria (Hyvärinen et al., 1991). It is therefore recommended that plates should be incubated at 20-25oC and examined daily for several days (Commission of the European Communities-Directorate General for Science, Research and Development, 1993).
References
Hyvärinen A.M., Martikainen P.J., Nevalainen A.I. (1991). Suitability of poor medium in counting total viable airborne bacteria. Grana 30, 414-417.
Commission of the European Communities-Directorate General for Science, Research and Development, Report No.12, Biological Particles in Indoor Environments, Brussels, Luxembourg, 1993.
L91-94 – please improve the description of ozone capturing for measurement. What was the absorption wavelength used. How was the calibration curve done?
The sample collection included passing 10 dm3 of the air at a rate of 1 dm3/min through a two scrubber containing 10 cm3 of a 1% KI solution. The determination of ozone concentration in the sample included 5 cm3 of the absorbing solution transferred to a colourimetric tube, next 0.5 cm3 of 0,02% dimethyl-p-phenylenediamine hydrochloride is added and mixed. After 15 minutes, the colour of the tested solution was compared in visible light (l=550 nm) to the reference prepared in accordance with the scale of standards.
L112 – “…and during the weaker sterilization..” – What did you mean? Did you use different voltage setup that influenced the sterilization intensity? – please improve
“weaker sterilization” means that we used there only one EWG device while the space of this room was larger compared to other rooms. The text is now changed:
“Table 3 shows that in the larger patient`s room (PRL) and during the weaker sterilization process (only one EWG device was used while the space of this room was larger compared to other rooms) the decrease of the concentration of bioaerosol in the hospital in Zabrze was not such strong as in the hospital in Sosnowiec.”
L120 – “Significantly lower results…” a) if lower bacteria/fungi concentration than I do not agree. B) may be lower sterilization efficiency.. – please improve. Moreover, significantly lower are results that are presented in Tab 3 than in Tab 4, comparing to those in Tab 2.
Thank you for finding this error. This sentence was amended as follows: "A significant reduction in bioaerosol levels due to the work of the EWG device was also received in the clinical waiting room (Table 4). It should be noted that after four hours of sterilization the concentration…..”
In figure 1 authors proved exp. decay in WR in Sosnowiec. What about two remaining rooms? Did they confirm the theory? Please refer to other cases.
The studies, the results of which are shown in Fig. 1, were carried out in the medical waiting room, where many people coming but for a relatively short time (1-1.5 hour). On average, there were 12-15 people at the same time, not very seriously ill. This allowed us to conduct tests for over 6 hours without patient protests. Meanwhile, the rooms in both hospitals had seriously sick patients who did not agree for the continuation of our study longer than 3-4 hours. Unfortunately, after several hours of operation of the EWG device, intensive sparking from the electrodes followed, which was additionally caused ozone emission. For this reason, we could continue our studies there no longer than 4.5 hours.
Again, we would like to express our appreciation to you and the reviewers for your efforts and helpful comments. Please find the revised version of our paper enclosed.
Yours sincerely,
Anna Mainka
Round 2
Reviewer 1 Report
L57 'on the other hand' is a slang like term please make more academic
L62/63 sentence is awkward
L63/64 get rid of the end sentence of this paragraph not needed
L81 what do you mean by 'in most cases'?
L137 'not such strong' you mean 'not as concentrated'?
Author Response
December 3rd, 2019
Title: "The study of the sterilization of the indoor air in hospital / clinic rooms by using the electron wind generator"
Manuscript ID: ijerph-637282
Dear Reviewer 1,
Thank you for your message with comments on our manuscript. We appreciate the responses and helpful suggestions for improving the presentation of our study. We have followed the comments and marked them in the text (yellow colour).
L57 'on the other hand' is a slang like term please make more academic
We changed on the other hand into alternatively.
L62/63 sentence is awkward
We changed the sentence:
This paper describes our studies, started in the period 2001-2009 and next continued in the last ten years.
into
The first stage of our studies on the subject started in 2001-2009 and have been continued in the last ten years.
L63/64 get rid of the end sentence of this paragraph not needed
We deleted the end of the sentence.
L81 what do you mean by 'in most cases'?
It should be:… at least 3 samples were taken.
L137 'not such strong' you mean 'not as concentrated'?
Yes.
Again, we would like to express our appreciation to you and the reviewers for your efforts and helpful comments. Please find the revised version of our paper enclosed.
Yours sincerely,
Anna Mainka